# Moving beyond Titers

**DOI:** 10.3390/vaccines10050683

**Published:** 2022-04-26

**Authors:** Benjamin D. Brooks, Alexander Beland, Gabriel Aguero, Nicholas Taylor, Francina D. Towne

**Affiliations:** 1Department of Biomedical Sciences, Rocky Vista University, Ivins, UT 84738, USA; 2Inovan Inc., Fargo, ND 58103, USA; 3College of Osteopathic Medicine, Rocky Vista University, Parker, CO 80112, USA; alexander.beland@rvu.edu (A.B.); gabriel.aguero@rvu.edu (G.A.); nicholas.taylor@rvu.edu (N.T.); ftowne@rvu.edu (F.D.T.)

**Keywords:** vaccine non-responsiveness/poor durability/longevity, titers, immune repertoire sequencing, immunogenomics, HLA typing, B cell epitopes, epitope binning, immunogenicity, vaccine efficacy, COVID-19, vaccine development

## Abstract

Vaccination to prevent and even eliminate disease is amongst the greatest achievements of modern medicine. Opportunities remain in vaccine development to improve protection across the whole population. A next step in vaccine development is the detailed molecular characterization of individual humoral immune responses against a pathogen, especially the rapidly evolving pathogens. New technologies such as sequencing the immune repertoire in response to disease, immunogenomics/vaccinomics, particularly the individual HLA variants, and high-throughput epitope characterization offer new insights into disease protection. Here, we highlight the emerging technologies that could be used to identify variation within the human population, facilitate vaccine discovery, improve vaccine safety and efficacy, and identify mechanisms of generating immunological memory. In today’s vaccine-hesitant climate, these techniques used individually or especially together have the potential to improve vaccine effectiveness and safety and thus vaccine uptake rates. We highlight the importance of using these techniques in combination to understand the humoral immune response as a whole after vaccination to move beyond neutralizing titers as the standard for immunogenicity and vaccine efficacy, especially in clinical trials.

## 1. Introduction

To date, vaccination has proved to be a cost-effective method to reduce the spread of infectious diseases and decrease morbidity and mortality associated with communicable diseases [1]. A single influenza season can have an economic impact of nearly USD 2 billion in the United States, and the COVID-19 pandemic showed that the economic impact associated with easily transmittable emerging diseases could be even more extensive [2,3]. Systematic reviews show that costs associated with treating those most in peril of infection, such as elderly populations, can be decreased with nationwide vaccination programs [4,5]. Large-scale vaccination campaigns in children can reduce mortality from some of the most dangerous infections. Disability-adjusted life years associated with many infectious diseases in children and adults have decreased since the late 1800s, primarily due to vaccinations efforts [6,7]. As the need for vaccinations continues, more thought is being given to developing more safe and cost-effective vaccines, making the evaluation of antibody and T-cell repertoires more targeted, and discovering how new technologies can assist in vaccine development. There are numerous examples of successful vaccine development; however, we propose that emerging technologies be utilized to ensure that current vaccines are even more safe and more effective. Potentially, with the adoption of these technologies into existing vaccine discovery and development workflows, the next generation of vaccines can be developed for difficult pathogens.

On average, vaccines require ten years to create a final vaccine product, from research and development to preclinical studies to licensing and can cost more than USD 1 billion [8,9]. COVID-19 vaccines were developed in less time but at a much greater cost and risk of failures that are likely not applicable with most vaccines. The significant vaccine development costs and time requirements underscore the need for new technologies to streamline the discovery and development processes. At the same time, they also speak for a more substantial public investment into preventative medicine. Public investment is worthy of consideration to supplement major pharmaceutical companies’ required profit motive that prevents the development of less profitable but more impactful vaccines.

Microorganisms have evolved numerous immune evasion mechanisms [10,11,12,13,14,15,16]. Several common pathogens obstruct successful vaccine development by interfering with host immune responses through various evolutionary strategies to immune senescence of the adaptive immune response in these patients [17]. While these pathogens make vaccine development challenging, emerging technologies may allow for the creation of vaccines that mitigate at least some of the problems presented by these complex pathogenic mechanisms.

We hypothesize that vaccines to difficult diseases can be developed, and current vaccine safety and efficacy can be improved with emerging molecular techniques. We propose improving vaccines and vaccine development with additional information beyond neutralizing titers, detailed in Section 3 of this manuscript. As a note, the discussions in this article are intended and presented as ways to improve on the historic successes of vaccines. The discussion in this article cannot and should not be interpreted to represent any relationship or association with anti-vaccination viewpoints.

## 2. Vaccine Effectiveness

Determining vaccine efficacy/effectiveness is multifactorial and not completely characterized. Potential factors for decreased effectiveness include decreased opportunities for natural exposure to pathogens, pathogenic evolution, and route of administration [10,11]. However, vaccine efficacy/effectiveness frames vaccine regulatory approval. A vaccine failure occurs when a person fails to generate a protective or therapeutic response even though they received a vaccine to prevent the disease. It is important to note that vaccine failures in this discussion occur in individuals not general populations. Thus, characterizing and overcoming individual vaccine failures could improve a vaccine’s overall efficacy.

Primary failures, or vaccine non-responsiveness, occur when a vaccinated individual fails to produce an initial immune response to the vaccine [12]. An example of this classification is the Measles, Mumps, Rubella (MMR) vaccine, where vaccine non-responsiveness occurs in 2–10% of healthy individuals. While live-attenuated vaccines have a lower prevalence of primary vaccine failures, vaccine non-responsiveness occurs to some extent in every vaccine humans have created thus far [13,14].

In contrast, secondary vaccine failure, or poor vaccine durability (i.e., limited immunological memory or durability), occurs when a vaccinated individual produces an adequate immune response initially, but the humoral immunity generated by that response diminishes over time [12]. As an important note, boosters overcome limited durability, but uptake/adherence problems arise. The mumps outbreaks that started globally in 2018 were most likely due to secondary vaccine failure, as seroconversion after MMR vaccination was greater than 95% after one dose and nearly 100% after two [15]. If the 2018 mumps outbreaks were due solely to declining effectiveness and decreasing antibodies, the outbreak would be continuous. Limited vaccine durability is more commonly associated with subunit and toxoid vaccines such as the Diphtheria, Tetanus, and Pertussis (DTaP) vaccine. Some evidence suggests that durability is associated with antigen properties (Kennedy et al.). Toxoid vaccines may also fail due to “exposure threshold”; despite having adequate protective antibodies, a sufficiently large exposure to the toxin can overwhelm the developed response [12]. This phenomenon can also occur with high challenge doses in viral and bacterial infections if immunity wanes. Figure 1 visually demonstrates these classifications.

A third type of vaccine failure classification exists that differs from current classifications of vaccine non-responsiveness (Type #1) and poor persistence or durability (Type #2). In this classification, the vaccine elicits a strong, quantifiable immune response as designed but fails to protect from infection by mounting a humoral response that involves non-neutralizing antibodies. Following the previous naming convention, this tertiary failure is different from primary and secondary because it identifies problems with efficacy rather than just the quantifiable humoral response. When this occurs, antibody titers may appear to show an appropriate response against the pathogen even though the present antibodies are not protective. Unlike vaccine non-responsiveness (Type #1) and poor persistence (Type #2), this efficacy classification is not due to the immune system either failing to create a response or having a waning response, but an inability of the immune system through the vaccine to generate antibodies that provide correlates of protection. Long-term vaccine efficacy is multifactorial, including the generation of neutralizing antibodies, as well as proliferating immune memory [16]. Memory B cells generated in response to vaccination exist in the absence of antigen and only produce antibodies when prompted to differentiate into plasma cells after re-exposure to said antigen. The type of vaccine significantly impacts memory: compared to polysaccharide vaccines, protein or conjugated vaccines recruit helper T-cells and induce germinal center proliferation of plasma cells or memory B cells [16].

Regulatory agencies require measuring “neutralizing” titers during trials as a measure of immunogenicity. Using neutralization titers is problematic in several ways and begs for a more sophisticated, patient-centered assessment. First, variants and genetic diversity can affect neutralization assays, as seen in COVID [17]. Second, neutralizing titers in pathogens may be an oversimplification of neutralization (e.g., HSV). Many neutralizing epitopes may be helpful or even required to achieve neutralization [18]. Lastly, neutralization may be generated uniquely by patients. For example, in Herpes Simplex Virus, the data indicated that neutralization may be differentially achieved between individuals [18].

The endpoints for immunogenicity during trials are usually titers. ELISAs are commonly used to measure titers for immunogenicity/efficacy, especially in early development and clinical trials; however, ELISAs commonly do not provide the neutralizing capability of the humoral response. In some vaccines, ELISA has been shown to have an unsatisfactory correlation with antibody function, for example, as seen in some cases of mumps due to cross-reactivity of antibodies against parainfluenza antigens [19]. In addition, standardization of what titer ratio constitutes an effective vaccine may also produce false positives. For example, seasonal influenza vaccines are based on hemagglutinin and neuraminidase. Antibodies against hemagglutinin are considered neutralizing, and antibodies against neuraminidase lessen the severity of the illness [20]. Hemagglutinin antibody titer in the range of 1:32 to 1:40 is considered protective; however, this only represents a level at which approximately 50% of the population will be protected. Furthermore, antibody titer levels less than 1:32 to 1:40 did not predict protection against the influenza virus [21]. In order to assess functionality, additional tests are generally required. Moreover, these tests are generally not performed due to the time and cost associated with the tests.

We hypothesize that by developing high-information tests that identify both quantifiable antibody titers and the neutralization of the pathogen, i.e., efficacy, we can develop more vaccines against problematic pathogens, improve the effectiveness of vaccines of current vaccines, and clinically identify at-risk patients for improved safety. We propose moving beyond titers to high-information formats while maintaining the current cost and rigor of ELISA and other simple immunoassay formats.

## 3. New and Emerging Technologies

As the traditional model of vaccines to combat infectious diseases has been criticized, a new generation of promising technologies for vaccine development has emerged [22]. Here, we outline three emergent technologies used that have the potential to improve vaccine development against difficult pathogens. We have focused on these because, when used together, these techniques provide synergy in overcoming many of the vaccines’ failures detailed above.

### 3.1. Epitope Characterization

Epitope characterization from vaccinated animal or patient samples, including clinical trials, can be used to improve vaccine development by providing more detailed molecular information. Epitope binning is an analytical technique by which antibodies generated by vaccinated individuals can be organized into categories determined by the epitopes to which they bind, both linear and conformational. Known monoclonal antibodies can compete with sample antibodies for binding to a target of interest. If the monoclonal antibodies and sample antibodies competitively bind the antigen, they will be “binned” together, given their ability to bind the same epitope [23,24]. The incorporation of epitope binning into vaccination development streamlines the empiric selection of epitope targets [25]. As shown recently in an analysis of herpes simplex vaccine candidates in animal studies and HerpeVac clinical trial, epitope binning can identify key epitomes and epitope profiles, both linear and conformational [25].

Epitope binning is currently being incorporated into some vaccination development processes [26,27,28]. Serum from vaccinated subjects can be incubated and allowed to form complexes with the antigen of interest. The serum antibody–antigen complexes can then flow over a biosensor chip with printed monoclonal antibodies to known epitopes of the antigen [26]. Surface plasmon resonance (SPR) analysis of the biosensor chip can reveal whether or not each antibody pair binds competitively to the same epitope [29]. Figure 2 demonstrates the workflow. SPR analysis is beneficial as it uses very little sample, though in some cases, biolayer interferometry (BLI) detection may increase the number of analyzed analyte/ligand pairs [30]. These processes reveal the epitopes with which each sample’s antibodies interact. In vitro neutralization assays and animal studies can further elucidate which epitopes correlate with immune evasion by the pathogen of interest [26]. Vaccine design could then be enhanced by selecting the epitopes that confer the most robust immune protection.

As noted above, current measurements of vaccine efficacy rely almost entirely on antibody titers. These tests provide quantitative information about an individual’s immune response but reveal very little qualitative information about how effective that immune response is [15,31,32,33]. Epitope binning can measure how and where a vaccine successfully generates an antibody response to the intended antigen. The organization and characterization of epitopes can identify highly conserved domains in pathogens that demonstrate high levels of antigenic variation, which would allow for a more targeted vaccine development process [34]. Epitope characterization can also reveal which antibodies have the largest intended biological effect and can be used to ensure that vaccines induce antibodies specific to epitopes with the most effective responses [35]. Epitope characterization could also allow for better analysis of existing vaccines by identifying which epitopes correspond with vaccine failures and which correspond with successful immunological protection [26]. In the past, vaccine epitope characterization may have been limited to linear epitopes, but as shown in Herpevac analysis, conformation epitopes can also be identified [18]. Further, the study showed that a complex profile of epitopes could provide protection and provide additional viral entry characterization [25].

### 3.2. Immunogenomics and HLA Typing

Vaccinomics integrates immunogenetics/immunogenomics with both immune profiling and systems biology. Although vaccinomics has been around for some time, one wonders if we have moved beyond the limited “isolate, inactivate, and inject” approaches. Furthermore, while these approaches have produced strong results, characterizing genetic/genomic obstacles to difficult pathogens has the potential to have a significant impact on public health [36]. The generation of a robust, durable, and protective immune response to a pathogen requires a complex series of biological processes mediated by genetics [36]. The genes involved include innate responses, antigen processing and presentation, immunoregulation, and many immune-modulating pathways.

Maybe the most impactful are the polymorphism in HLA haplotypes which lead to a wide array of antigen presentation and varying levels of effective immune response [37]. In peptide vaccine candidates for measles/mumps and hepatitis, the immunologic response is affected by the variants of HLA that the individual has. Specifically, HLA-DRB1 is most implicated in the immune response to several vaccines [38]. Different HLAs may bind with different avidity to peptides in vaccines or may bind to different epitopes altogether. For example, HLA-DRB1*03, HLA-DPA1*0201, and HLA-DRB*0701 are associated with poor immune responses in measles and hepatitis B vaccines, respectively [38].

Genetic variations in HLA also affect immune responses against natural pathogens in a similar manner. Not only are HLA haplotypes associated with poor immune response, but some haplotypes are associated with a more robust response as well. Conjugate vaccines such as Hemophilus influenzae type b and capsular group C meningococcal are bound to tetanus toxoid. Including the tetanus toxoid vaccine, these three vaccines have a more durable immune response associated with specific HLA classes: HLA-DQB1*0201, HLA-DRB1*0301, HLA DQB1*0602, and HLA DRB1*150 (69). In the 2020–2021 pandemic of COVID-19, initial data showed that HLA-B*46:01 has the fewest predicted binding peptides to SARS-CoV-2 proteome, which causes COVID-19 syndrome [39]. While theoretical, this may represent a genetic component correlating to more severe disease. In contrast, HLA-A*25:01, HLA-B*15:03, and HLA-C*12:03 were the best presenters of novel coronavirus peptides [39]. HLA-B*15:03 also presented peptides shared among human coronaviruses and might signal that this gene variation may confer T-cell-mediated protection against not only COVID-19 but other coronaviruses that cause SARS or MERS as well. This predictive genetic information can determine who is more likely to have a severe course of disease and aid in who needs a vaccine. More importantly, using HLA haplotypes to determine which bound epitopes are the best targets for vaccine development could improve vaccine safety and efficacy.

Due to the vast array of heterogeneity in HLA haplotypes, novel vaccines need to account for the genetic variability of HLA, or testing needs to be carried out to assess for HLA compatibility [40]. Personalized vaccines based on an individual’s genetics are a multi-faceted process. Host polymorphisms are significant factors in inducing an appropriate immune response to vaccines. Using this immunogenomic information may aid in designing vaccines that avoid genetic restrictions. In one study, researchers isolated vaccinia and measles-related peptides from naturally exposed HLA molecules associated with a poor immune response to vaccines [37]. They note that soon, these peptides could be used to create vaccines that avoid HLA polymorphic restrictions [37]. The advent of immunogenomics, which also examines additional genetic polymorphisms such as single-nucleotide polymorphisms, can introduce safer and more effective, personalized vaccines. In addition to using an individual’s genetic makeup, vaccines can also be more effective by using methods to find epitope-based peptides that more effectively bind to T cells based on an individual’s HLA type to modulate immune response [41].

### 3.3. Antibody Repertoire Sequencing

Antibody repertoire sequencing to a vaccine is a technique used to characterize the host’s antibody repertoire and other humoral response parameters to a vaccine [42,43]. Historically, first-generation sequencing has been used to create fractured lengths of a template strand using labeled dideoxynucleotides. Then, the complementary sequence is read on a polyacrylamide gel or software to interpret labeled dideoxynucleotides (see Figure 3) [44]. Major challenges often hinder antibody repertoire sequencing. The utility of these sequences is based on a more complete understanding of the humoral immune system and the evolution of the B cell from pre-B cell to plasma cell [45]. Only after the completion of B cells activation, including somatic hypermutation, can the interpretation of the library of B cells responses be identified. Otherwise, sequence chaff from pre-B cells and naïve B cells make identification of mature B cells’ DNA sequences very difficult, something many ignore over the convenience of having massive quantities of data from NGS of PBMCs and then rely on algorithms to separate the chaff from the few real grains [46]. This shotgun approach, however, makes interpretation of a particular immune response very limited, especially when linkage of heavy and light chains is not preserved. Often, the sequences lack context and do not have sufficient information for the entire variable region, thus making synthetic antibodies that may not necessarily reflect their original sequence. This problem is amplified by the use of A.I. and computational methods, creating novel entities that may not be biologically compatible with the human immune system [47]. Lastly, the synthesis of antibodies from NGS sequences can be cost-prohibitive when it has to be done for hundreds or thousands of individual antibodies. Thus, most of the antibody repertoire sequencing so far has focused on obtaining limited sets of antibodies, not whole repertoire analysis in response to disease or vaccination.

The diversity of antibody sequences can be examined through the generation of big data (e.g., machine learning). Similar antibody sequence patterns can be grouped together to create a repertoire of sequences that correspond to strong antibody affinity against specific pathogens [48]. This repertoire can be further enhanced by analyzing mutations in antibody sequences that correlate with high specificity against a pathogen of interest [48,49]. Antibody libraries are used to create antibody repertoires from individuals recovering from infections [50]. The antibodies with neutralizing epitopes are identified, and epitopes that have little effect when bound by an antibody can be recorded [50,51]. This allows for researchers to generate synthetic antibodies with antibody sequences that contain high specificity for molecular patterns of pathogens of interest as a method of passive immunization against a pathogen, with the caveat that synthetic antibodies have potential issues of their own in terms of compatibility with the human immune system [52,53]. Researchers are also able to generate key information about how antibodies interact with pathogens. This information can be used to better analyze antibody titers, improve comparison of host antibody response against a pathogen by analyzing antibody quantity along with antibody repertoire produced by different persons against the same pathogen and assist in the creation of vaccines that have higher efficacy of binding to certain neutralizing epitopes [54]. The ability to distinguish between antibodies is very complicated because their derivation from a limited set of genetic fragments leads to millions of often only slightly different proteins, and what makes some effective and others not, can require a significant amount of investigation. Doing so for thousands or millions of different ones quickly creates a jumbled mess of data, making the data processing and interpretation a bottle neck for this promising field at this junction.

The creation of antibody repertoires can be used to improve vaccines by promoting a targeted humoral immune response against neutralizing epitopes conserved across mutated pathogen genotypes [55]. Antibody repertoires can be determined before vaccination to measure pre-existing immunity [56]. Examining pre-existing immunity can help to establish the degree of “original antigenic sin”, where one’s existing antibody repertoire from previous vaccination or disease exposure dominates in response to a subsequent challenge, which can impede the generation of novel antibodies to different antigenic epitopes in response to that challenge [56,57]. Antibody repertoires can also be evaluated between vaccinated and naturally infected persons to compare neutralizing immunoglobulins expressed and determine which immunoglobulins persist over time; however, at the time of writing, there is yet to be anyone who can interpret these different repertoires [57]. Knowing one’s pre-existing immunity, immunoglobulin repertoires that have persisted over time in response to vaccination in comparison to infection, how that repertoire changes over time, and which of the neutralizing immunoglobulins persist offers valuable information for vaccine development. With more research into what the immune repertoire data describe, these data can eventually be used to design more personalized vaccines. Based on analysis of one’s immune repertoire, a vaccine that promotes neutralization to antigens that may be missed in a mass-produced vaccine can be utilized, conferring better individual protection [58]. Libraries of antibody repertoires can be used in conjunction with existing information on effective pathogen epitopes purposed for vaccines to create smarter vaccinations that produce host responses with more effective neutralizing immunoglobulins across more lines of mutating pathogen genotypes [55]. Maybe even more important would be the ability to identify antibody-dependent enhancement (ADE) possibilities that could improve some vaccines’ safety. The beginnings of this have been seen in the development of COVID and COVID vaccines [59,60,61].

## 4. Advancement Opportunities

Vaccines have reduced efficacy for a multitude of different reasons. Here, we propose extending the current vaccine response classification to include a robust humoral response that is not neutralizing (i.e., tertiary vaccine failure). For example, the mumps virus component of the MMR vaccine has been shown to produce a non-neutralizing humoral response despite strong positive titers in some individuals. This is partly due to titers having cross-reactivity with parainfluenza antigens [19]. Vaccination against HSV-2 has proven problematic in the past due to failure to produce specific antibodies against immune evasion domains on HSV-2 envelope glycoproteins C, D, and E (gC2, gD2, and gE2, respectively) [27,62]. Once these glycoproteins have been identified, specific epitopes can be identified that are neutralizing through epitope binning. This approach could bring a renaissance in vaccinology by determining epitopes and corresponding antibodies that produce the most effective vaccines. In addition, with the combined use of antibody repertoire sequencing and HLA typing, we could identify problematic genotypes that require additional vaccine design consideration.

The recent emergence of H1N1 influenza, Ebola, Zika, SARS-CoV-1, and most notably, SARS-CoV-2 demonstrate the need for rapid production of safe and effective vaccines to combat epidemics and pandemics [63]. The Zika and SARS-CoV-1 epidemics ended before vaccines could be completed [63,64]. Rapid production of vaccines for Zika and SARS-CoV-1 failed mainly because they were assessed mainly by titers, i.e., general binding affinity instead of targeting specific epitopes that confer robust immunological protection [65,66,67]. By selecting for therapies based on general affinity, vaccines that produce antibodies that bind weakly to areas of high biological function may be passed over in favor of “dead-end” vaccines that generate antibodies that bind strongly to areas of little biological relevance. In contrast, epitope binning allows for early recognition of vaccines that yield antibodies that bind to regions of biological relevance. Targeting known epitopes early in the vaccine development process significantly reduces the cost and time of vaccine design for emergent diseases by eliminating “dead end” vaccines from the development pipeline [68]. For example, recent studies used BLI and epitope binning to discover four distinct epitopes targeting the spike RBD [69].

With the emergence of COVID-19 and global variants of concern (VOCs), the push to develop and combine technologies has never been more important. Patients ill with COVID-19 may develop measurable antibody titers and neutralizing antibodies, but this does not identify individuals susceptible to severe disease progression [70]. As shown in influenza, antibody titers do not always correlate with—and cannot always predict—strain-specific vaccine efficiency [71]. Modern technologies, such as the mRNA vaccines and viral vector vaccines, protect against COVID-19 and VOCs and boost immunity in those previously infected [72]. Other technologies, such as HLA-typing, show promising results in vaccine research and design based on the distribution of MHC alleles in a population to estimated coverage with a particular set of T-cell epitopes [73]. Antibody repertoire sequencing will likely play a role in identifying “at-risk” individuals in the future.

In addition, emergent disease vaccines require a strong safety profile to avoid hesitancy. These techniques can provide additional layers of evidence for safety to provide reassurance. These techniques can identify individuals who respond favorably and those who respond poorly to developed vaccines, particularly when utilized in conjunction with other vaccines (e.g., COVID and influenza administered simultaneously). Just as pharmacogenetics has de-risked specific pharmacological product developments, we anticipate that these techniques will decrease adverse events and increase successful immunity by identifying individuals who respond well to particular vaccines [74,75]. As noted in the introduction, microbes have sophisticated immune evasion mechanisms. Again, as the HerpeVac trial analysis by Cairns et al. showed, these techniques will provide additional information to engineer vaccine candidates that address these mechanisms [25]. In the HerpeVac trial analysis, the brown community, the elbow of a flexible arm region that protected the nectin-binding region, was shown to generate antibodies with fast off-rates reducing or eliminating virus neutralization [25]. This information provides insight into the immune evasion mechanisms of HSV and potential points to address in vaccine design. As we see future COVID-19 VOCs and other emerging diseases, marrying the molecular approaches will be critical. Movement towards an integrated approach, non-reductionist, is needed. There could not be a better time to merge multiple techniques that will allow quicker and more personalized advances in the future.

Rapid vaccine development could facilitate epidemiology by providing information about the immune response to both the pathogen and the vaccine. We are seeing the contribution of these techniques to reduce the spread of certain strains of disease and assist in creating vaccines against previously thought-to-be vaccine-resistant pathogens. Production of a more effective CMV vaccine is now possible; a specific immunoglobulin to a neutralizing epitope on the virus can now be characterized, even though it is not the most abundant antibody in a titer [76]. Hypervariable pathogens such as HIV, where vaccination poses a daunting task, are more approachable, although still challenging, with the ability to sequence humoral antibody response and work backward to create a vaccination [77]. Additionally, in less than a week, scientists were able to identify the causative agent of a lower respiratory infection, the novel coronavirus, in January 2020 through the use of real-time PCR, next-generation sequencing, and analysis against existing data bank sets [78]. These promising studies raise hope that the next generation of vaccines can contribute to a measurable decrease in the spread of pathogens and a more rapid response against newly emerging pathogens.

Most vaccine candidates fail to reach the market after ten years of development [79]. The vaccines that do not make it to market occur largely due to technical difficulties that arise from targeting biologically complex pathogens and from the massive cost of clinical trials [9]. However, the techniques we have described can alleviate these economic burdens by at least partially de-risking development, particularly in the early stages. Incorporating epitope binning early in the discovery process identifies ideal targets for development and prevents vaccines that induce “dead end” antibodies and noted safety concerns from reaching more expensive phases of development [68]. Additionally, the immense cost of safety and efficacy trials for vaccines can be reduced by decreasing the risk of failure by utilizing vaccinomics [75,80]. HLA typing and B cell sequencing provide information that can identify ideal candidates for vaccinations and eventually identify participants that are potentially genetically predisposed to experience adverse events. These techniques can reduce the total cost of clinical trials, just as utilizing pharmacogenomics has done for pharmaceutical development [81].

As COVID and COVID variants have demonstrated, response to a vaccine is exceptionally personalized. The mantra in cancer research has been personalization, and in the future, vaccinology should consider adopting this mantra. The cancer field recognized that cancer is also highly personal and uses genomics to diagnose and treat the individual. Additional information noted in Section 2 could also be applied effectively to vaccinology.

## 5. Conclusions

Vaccines have proven to be one of the most significant developments of the modern world. By reducing the spread of infectious diseases and decreasing morbidity and mortality, vaccines have improved the quality of life of millions worldwide [1]. Compared to other medical advances, vaccines are also cost-effective to keep global populations healthy [4,5]. Novel technologies allow vaccinology to progress to a new era of even more effective vaccines. We now know that there is a way to move vaccinology to incorporate a patient’s unique immune system and genetics into producing a vaccine. We can determine the most biologically significant epitopes and use those to create better vaccines. In addition, we now have the technology to test for more specific antibodies and T cells to ensure adequate immunity. We call on companies and governments across the globe to invest in these innovative technologies together early in development and clinical trials.

While most vaccines do not require an individual to have lab work beforehand, HLA typing would require an extra step in vaccine administration. In the current health care climate, this will likely lead to decreased uptake due to increased cost and time associated with multiple doctor visits. Antibody repertoire sequencing is fast; however, it is also costly, data-intensive, ambiguous, and highly technical. As such, proper antibody repertoire sequencing is not practiced yet, and the advanced skill set would have to be standardized and taught to many researchers to use the technique on a large scale. Epitope binning is initially cost-prohibitive; however, the potential to utilize a single biosensor chip for multiple analyses may prove to be cost-productive [26]. While these are significant obstacles, we believe that these technologies will only become more conventional and thus less expensive to use. In addition, we believe that the expense of these technologies may be offset by a reduction in vaccine development costs and decreased time to market. The additional testing and medical visits that HLA typing requires may seem unfavorable but may only be needed once in a lifetime. Additionally, the implementation of HLA typing only needs to occur post-vaccination if immune responses are insufficient. In addition, immunogenetics and genomics can be applied to provide additional insights. Overall, medicine is trending to more individualistic care, and the implementation of pre-and post- vaccination laboratory work will likely become the new standard.

As noted above, pathogens use natural selection and mutation to make vaccine development challenging. These emerging tools could allow for the creation of vaccines that could overcome many of the problems presented by these complex pathogenic mechanisms and mutations. At bare minimum, as noted in several examples above, viral mechanisms could be further characterized that inform different solutions. The bottom line is that with the extensive dataset generated through the use of these tools, science can learn microbial mechanisms in our war against them and develop vaccines to fight them.

The main challenge with this approach, however, is cost. To characterize vaccines at this level would likely increase both the R&D as well as the individual cost of vaccination. For some vaccines, this would not meet the cost/safety/efficacy trade-off requirements to bring the vaccine through to market; in current health care markets, this approach may require government subsidization to address the market failures of vaccines. For other vaccines where the disease burden is high, especially in high-risk groups and where the viruses pose challenges to vaccine development such as the Herpesviruses or HIV, the cost of disease burden may justify protection and reduction in disease burden costs. As with numerous other pharmacological interventions, lower-income countries’ economics may require additional supplemental or economic incentives.

Here, we propose moving beyond titers in characterizing vaccine responses. We have delineated several emerging methods to develop tests that allow for high-information vaccine characterization results. HLA typing, antibody repertoire sequencing, and epitope binning represent new frontiers in vaccinology. The techniques used together early in development and trials could reduce the gaps in protection (i.e., vaccine failures). The costs associated with treating preventable infectious diseases, a significant death toll from these diseases in developing countries, and the emergence of new pandemics justify using these emerging technologies. Their use in both the characterization of vaccine response and vaccine development has the potential to revolutionize the industry by producing better, faster, and more effective vaccines for pathogens that we do not yet have immunizations for or have imperfect vaccinations currently. We have come together numerous times before to improve vaccine development, and this attempt seems like a strong, logical next step.

## Figures and Tables

**Figure 1 vaccines-10-00683-f001:**
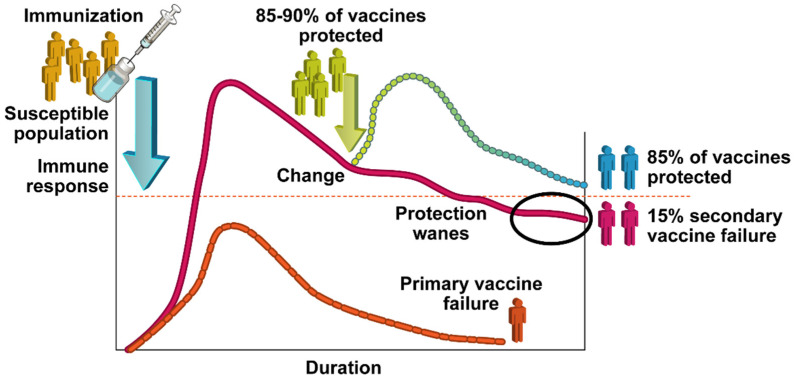
Representation of vaccine failures. After immunization, a humoral response is generated. A primary vaccine failure occurs when the immune response fails to generate sufficient protection levels, or titers (shown in orange). The immune response level sufficient for protection is shown as the dashed red line. A secondary vaccine failure occurs when protection levels wane.

**Figure 2 vaccines-10-00683-f002:**
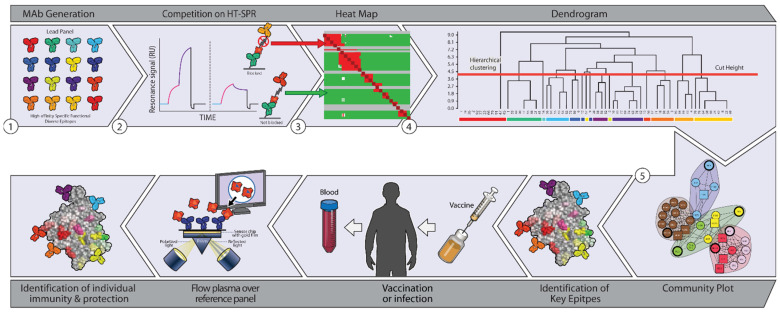
Steps in epitope characterization assays for characterizing the vaccine-induced humoral response. Step 1. Antibodies to microbial protein targets are generated. Step 2. Epitope binning assays are performed using monoclonal antibodies generated in step 1. Step 3. A heat map is generated from the epitope binning, or cross-competition assays. Step 4. Dendrogram from the heat map is generated. Step 5. A community plot is generated from the dendrogram, which groups antibodies by their epitope binding regions. Step 6. A panel of antibodies with broad coverage of all the epitopes on microbial targets is generated. Step 7. Patients are immunized, and plasma at time points are collected. Step 8. Plasma flows over the reference panels from step 6 on a biosensor where the bound epitopes are identified. Step 9. The patient’s humoral response by epitope is generated, identifying which epitopes have coverage. Neutralizing sites or profiles at the epitope level can then be identified by comparing them to neutralization assay results (not in a one-on-one relationship).

**Figure 3 vaccines-10-00683-f003:**
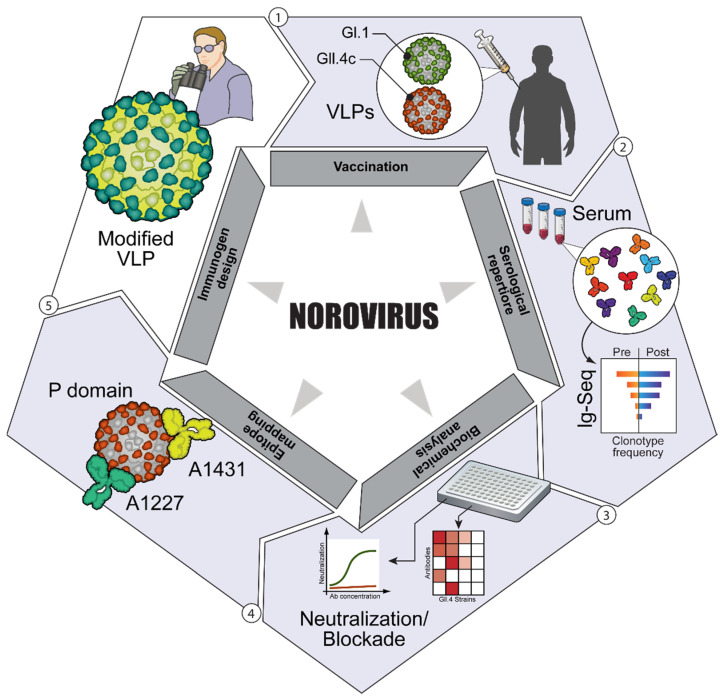
B cell sequencing steps. Step 1. Immunization is administered generating a humoral response. Step 2. PBMCs are collected, and DNA sequencing is performed on B cells. Step 3. Recombinant antibodies are generated and tested for neutralizing. Step 4. The epitopes of the recombinant antibodies are identified using epitope characterization and analyzed with neutralization data. Step 5. The vaccine is modified to improve humoral response.

## Data Availability

Not applicable.

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
