# Peer review of "Moving beyond Titers"

_vaccines, 2022, doi:10.3390/vaccines10050683_

Round 1

Reviewer 1 Report

Comments on the manuscript: Moving Beyond Titers 2 by Benjamin D. Brooks et al.

Protection against infection and or disease is an intricate collaboration between innate and adaptive T and B cell responses. Efficient protective antibody responses depend on T cell help, in particular at germinal centres. T cells are important components for protection against many diseases. i.e., patients with primary or secondary antibody deficiency syndrome and reduced or absent B cells can recover from COVID-19. SARS-CoV-2 specific T cell immune responses but not neutralising antibodies are associated with reduced disease severity in these patients. (Quinti et al. J. Allergy Clin. Immunol. 2020; 146. 211-213.e4 https://doi.org/10.1016/j.jaci.2020.04.013, Montero-Escribano et al., Mult Scler Relat Disord. 2020; 42.102185. https://doi.org/10.1016/j.msard.2020.102185). For vaccine clinical trials neutralising or bactericidal antibody titres are preferred over ELISA antibody titres to evaluate vaccine immunogenicity, T cell responses is not always evaluated, however as shown for COVID-19 vaccines this could be another correlate of protection. However, what dictates if a vaccine will be approved is its capacity of protect against infection and/or disease and dead.

  1. Authors state that “vaccine failures are classified as vaccine non-responsiveness (Type #1) and poor durability/persistence (Type #2). A third type of vaccine failure classification exists that differs from current classifications of vaccine non-responsiveness (Type #1) and poor persistence or durability (Type #2).In this classification, the vaccine elicits a strong, quantifiable immune response as designed but fails to protect from infection by mounting a humoral response that involves non-neutralizing antibodies”. However, during vaccine development process, the experimental vaccines (prototypes), which do not show protective capacity in animals and in humans, do not become vaccines. The “Type #3” failure exclude experimental vaccines to actually become vaccines.
  2. How these technologies could address the identification of epitopes composed of peptides and its associated carbohydrates?
  3. The manuscript presents an interesting viewpoint about the use of new technologies to improve vaccine discovery and improve vaccines in general focusing only in the antibody responses. However as explained above, protection against infection or disease is much more complex. Indeed, the technologies described in the manuscript can be of great help to identify the correlates of protection, which is one of the main challenges to design new vaccines, in addition, could importantly help to identify important components of a vaccine. Nevertheless, the manuscript addresses the problem only from the side of the antibody response and does not clearly discuss T cell and innate correlates of protection. Moving beyond titres is an attractive title, since in vaccine development field has been discussed from long time that antibody responses (in particular, neutralising or bactericidal antibodies) represents only a partial view of immunity. From this title was expected the description of technologies to help to identify correlates of protection beyond antibodies. In addition, discussion about protective vs non-protective antibodies is not new, and is quite accepted in the field the need of better ways to identify and induce protective antibodies. Taken together, the viewpoints expressed in the manuscript address only partially how these new technologies could help to improve vaccine discovery and development. Manuscript would importantly improve with a clear explanation of how these technologies can contribute to identify correlates of protection beyond antibodies.

Minor points:

Some references are misplaced i.e Ref 50 mentioned COVID, however manuscript subject is HIV-1. Reference 51 is also misplaced in the text lines 128 and 129.

In lines 264 and 265 is stated: “vaccines can also be more effective by using methods to find epitope-based peptides that most avidly bind to T cells to modulate immune response”. Epitopes does not bind T cells, please revise this concept.

Please revise line 276: “Only after the final stages of development, including somatic hypermutation, can the interpretation of this vast library of responses be identified”. Is not clear what authors want to say.

Author Response

Protection against infection and or disease is an intricate collaboration between innate and adaptive T and B cell responses. Efficient protective antibody responses depend on T cell help, in particular at germinal centres. T cells are important components for protection against many diseases. i.e., patients with primary or secondary antibody deficiency syndrome and reduced or absent B cells can recover from COVID-19. SARS-CoV-2 specific T cell immune responses but not neutralising antibodies are associated with reduced disease severity in these patients. (Quinti et al. J. Allergy Clin. Immunol. 2020; 146. 211-213.e4 https://doi.org/10.1016/j.jaci.2020.04.013, Montero-Escribano et al., Mult Scler Relat Disord. 2020; 42.102185. https://doi.org/10.1016/j.msard.2020.102185). For vaccine clinical trials neutralising or bactericidal antibody titres are preferred over ELISA antibody titres to evaluate vaccine immunogenicity, T cell responses is not always evaluated, however as shown for COVID-19 vaccines this could be another correlate of protection. However, what dictates if a vaccine will be approved is its capacity of protect against infection and/or disease and dead.

            We completely agree with the reviewer. We have addressed this below.

  1. Authors state that “vaccine failures are classified as vaccine non-responsiveness (Type #1) and poor durability/persistence (Type #2). A third type of vaccine failure classification exists that differs from current classifications of vaccine non-responsiveness (Type #1) and poor persistence or durability (Type #2).In this classification, the vaccine elicits a strong, quantifiable immune response as designed but fails to protect from infection by mounting a humoral response that involves non-neutralizing antibodies”. However, during vaccine development process, the experimental vaccines (prototypes), which do not show protective capacity in animals and in humans, do not become vaccines. The “Type #3” failure exclude experimental vaccines to actually become vaccines.

In the manuscript, we assume a minimum threshold of efficacy in a population for the vaccine such that vaccine failures occur in individuals not widespread in the general population. We have added discussion to clarify.

2. How these technologies could address the identification of epitopes composed of peptides and its associated carbohydrates?

As noted in lines 167-8 and 175-6, the technologies can accomodated peptide epitopes (linear). Carbohydrates are not accommodated with these technologies. This is definitely a limitation of the current technology, but it is also a limitation of the immune system (e.g. glycan shields).

3. The manuscript presents an interesting viewpoint about the use of new technologies to improve vaccine discovery and improve vaccines in general focusing only in the antibody responses. However as explained above, protection against infection or disease is much more complex. Indeed, the technologies described in the manuscript can be of great help to identify the correlates of protection, which is one of the main challenges to design new vaccines, in addition, could importantly help to identify important components of a vaccine. Nevertheless, the manuscript addresses the problem only from the side of the antibody response and does not clearly discuss T cell and innate correlates of protection. Moving beyond titres is an attractive title, since in vaccine development field has been discussed from long time that antibody responses (in particular, neutralising or bactericidal antibodies) represents only a partial view of immunity. From this title was expected the description of technologies to help to identify correlates of protection beyond antibodies. In addition, discussion about protective vs non-protective antibodies is not new, and is quite accepted in the field the need of better ways to identify and induce protective antibodies. Taken together, the viewpoints expressed in the manuscript address only partially how these new technologies could help to improve vaccine discovery and development. Manuscript would importantly improve with a clear explanation of how these technologies can contribute to identify correlates of protection beyond antibodies.

We completely agree with the reviewer that T cells are important. The review is focused on humoral immunity and combining the three technologies detailed in the manuscript to improve the identification of antibody protection. The T cells technologies are not synergistic with these technologies currently; moreover, they are not as developed, unfortunately. As a note, in a previous submission of this manuscript, we discussed T cells and were told we lacked focus so we removed the discussion to provide that focus

Minor points:

Some references are misplaced i.e Ref 50 mentioned COVID, however manuscript subject is HIV-1. Reference 51 is also misplaced in the text lines 128 and 129.

We have fixed these references. 

In lines 264 and 265 is stated: “vaccines can also be more effective by using methods to find epitope-based peptides that most avidly bind to T cells to modulate immune response”. Epitopes does not bind T cells, please revise this concept.

We have clarified the text.

Please revise line 276: “Only after the final stages of development, including somatic hypermutation, can the interpretation of this vast library of responses be identified”. Is not clear what authors want to say.

We have clarified the text.

Reviewer 2 Report

The authors nicely summarize vaccinology in vaccine development from an apparent medical and sociological perspective on how it can reliably produce safer and more effective vaccines. 

Unlike cancer treatment, prevention of infection with vaccines does not target individuals but rather the population. Indeed, it is technically feasible to manufacture vaccines based on the characteristics of individuals in advance, but when considering the acquisition of herd immunity, taking individual problems into account entails enormous costs. In this regard, I believe that how to find the most essential common denominator target is critical in ensuring maximum efficacy and safety at minimum cost.

As has been pointed out with the COVID-19 vaccine, even if only developed countries are prevented, it cannot be said that the pandemic has been conquered unless even developing countries are covered.

It has been reported that the smallpox vaccine developed in the 1970s is a fusion of conventional technology and vaccine mixes in the pursuit of safer and more effective (Eto et al. Biology, 2021, 10,1158). Based on the public health perspective of vaccines, the authors would also like to see suggestions on where to compromise price, safety, and efficacy.  

Other.

L195-198: It should also be noted that neutralizing antibodies and epitopes correlate, but not necessarily in a one-to-one relationship.

L396: "e.g.. ," is a typo.

Author Response

Unlike cancer treatment, prevention of infection with vaccines does not target individuals but rather the population. Indeed, it is technically feasible to manufacture vaccines based on the characteristics of individuals in advance, but when considering the acquisition of herd immunity, taking individual problems into account entails enormous costs. In this regard, I believe that how to find the most essential common denominator target is critical in ensuring maximum efficacy and safety at minimum cost.

As has been pointed out with the COVID-19 vaccine, even if only developed countries are prevented, it cannot be said that the pandemic has been conquered unless even developing countries are covered.

It has been reported that the smallpox vaccine developed in the 1970s is a fusion of conventional technology and vaccine mixes in the pursuit of safer and more effective (Eto et al. Biology, 2021, 10,1158). Based on the public health perspective of vaccines, the authors would also like to see suggestions on where to compromise price, safety, and efficacy.  

We concur with these concerns and have added discussion to this point in the conclusion.

Round 2

Reviewer 1 Report

Authors duly responded to all the comments raised in the review.

This manuscript is a resubmission of an earlier submission. The following is a list of the peer review reports and author responses from that submission.

Round 1

Reviewer 1 Report

The manuscript “Moving beyond titers” discusses how ELISA-based antibody titers and neutralization titers are not sufficient to characterize the humoral immune response to vaccines and that a more detailed analysis of the antibody response through sequencing and characterization of epitope specificities can potentially offer better correlates of protection. The authors also discuss the role of HLA polymorphisms in the variation of the immune response to vaccines. The authors suggest that these techniques can help to improve vaccine development and lead to a personalized approach to vaccination.

Unfortunately, the review lacks focus and the authors fail to demonstrate how the techniques that they have chosen could inform the development of more effective and safer vaccines. The authors exclusively discuss neutralizing antibodies and epitopes and ignore other functional aspects of antibodies that play a critical role in the protection against pathogens such as complement binding, ADCC, glycosylation, etc. The discussion of the genetic basis of the variability of the immune response is not clearly associated with the discussion of sequencing and epitope mapping. Furthermore, the authors focus this discussion exclusively on HLA polymorphisms and exclude other genetic variations that appear to contribute to quantitative and qualitative variations in the immune response to vaccines. As a whole, the manuscript fails to distill any new information from the reviewed literature.

A few specific comments:

  • Line 48-50. The authors should comment on the speed of development and approval of Covid-19 vaccines.
  • Table 1. It is not clear how many of these mechanisms of immune evasion challenge the development of vaccines as most of these occur after an infection has taken place. The authors omitted the most important mechanism by which coronaviruses avoid immune responses, .i.e, antigenic variation.
  • Line 75-76. “While live-attenuated vaccines have a more common prevalence…”. I don’t think that live attenuated vaccines are more common and fail to see how this is related to vaccine unresponsiveness.
  • 84-86. “vaccine persistence”. Should be duration of the immune response to vaccines. The duration of the immune response to tetanus is actually quite robust.
  • Line 87 – 89. ““exposure threshold,” despite having adequate protective antibodies, a sufficiently large exposure to the toxin can overwhelm the developed response.” This is not unique to toxoids and can also occur with viruses or other pathogens.
  • Line 104-108. “While a study showed no significant differences in pertussis and tetanus antibodies in children, another study showed a Th2-biased immune response and a decreased Th1 response, which usually mediates acute infections[35]. In addition, a study in healthcare workers vaccinated against hepatitis B showed high levels of non-responsiveness [35].” The authors base a large part of their review on other review articles. Reference 35 concerns the cost of vaccine development. The authors should focus their review on primary research articles.
  • Figure 2. “Neutralizing profiles can then be identified.” The authors should explain how steps 1 – 9 lead to neutralizing profiles.
  • 119 - . “The new vaccine platforms (e.g., mRNA) and the COVID-119 19 pandemic provide a unique opportunity to characterize memory in greater detail and 120 identify improved metrics for vaccine durability.” It is not clear why the mRNA vaccines would provide a unique opportunity compared with more traditional vaccine approaches.
  • Line 250. “In peptide vaccines such as measles/mumps and hepatitis”. Measles/mumps is a live attenuated vaccine.
  • Figure 3. It is not clear how epitope mapping of antibodies from patients immunized with subunit or killed vaccines can lead to modifications of the vaccine to improve the humoral immune response.
  • Line 448. Paragraph 4.3 has little to do with epidemiology or “Epidemiological advancement”.

Author Response

Here is our response to this review.  We appreciate the time and detail to improve the manuscript.

Unfortunately, the review lacks focus and the authors fail to demonstrate how the techniques that they have chosen could inform the development of more effective and safer vaccines. The authors exclusively discuss neutralizing antibodies and epitopes and ignore other functional aspects of antibodies that play a critical role in the protection against pathogens such as complement binding, ADCC, glycosylation, etc.

When writing this scoping review, we focused it on moving beyond titers for characterzing the humoral response. We would like to focus on the molecular and genomic aspects of the antibody response in relation to the profile of binding sites. We feel that the other aspects would detract from the focus. We have several cited papers that the authors have been associated where techniques have and are being used to improve vaccines (e.g. HSV).

The discussion of the genetic basis of the variability of the immune response is not clearly associated with the discussion of sequencing and epitope mapping. Furthermore, the authors focus this discussion exclusively on HLA polymorphisms and exclude other genetic variations that appear to contribute to quantitative and qualitative variations in the immune response to vaccines. As a whole, the manuscript fails to distill any new information from the reviewed literature.

We are not presenting new techniques. Instead, we are advocating for the combined use of these existing techniques to move to molecular and genetic characterization of the vaccine response for a broader use, especially in regulation. We are advocating for the use of these techniques in combination to improve vaccine development.

Line 48-50. The authors should comment on the speed of development and approval of Covid-19 vaccines.

We have added a discussion of the speed of the COVID vaccine development.

Table 1. It is not clear how many of these mechanisms of immune evasion challenge the development of vaccines as most of these occur after an infection has taken place.

We respectfully disagree with this point. First, several mechanisms are presented that occur before infection (e.g., HIV glycan shield, high mutation rates, genomic variation). In addition, the ability to inhibit vaccine development can occur after infection (e.g., inhibiting antigen presentation, blocking complement, blocking Fc receptors).

The authors omitted the most important mechanism by which coronaviruses avoid immune responses, .i.e, antigenic variation.

We have added the suggestion by the reviewer for coronavirus to the table.

Line 75-76. "While live-attenuated vaccines have a more common prevalence…". I don't think that live attenuated vaccines are more common and fail to see how this is related to vaccine unresponsiveness.

We have changed the wording of the sentence to clarify the meaning.

84-86. "vaccine persistence." Should be duration of the immune response to vaccines. The duration of the immune response to tetanus is actually quite robust.

We have changed the wording of the sentence to clarify the meaning. We appreciate the point that tetanus is quite robust but it does have limited durability. We agree that it isn't poor as we originally stated.

Line 87 – 89. ""exposure threshold," despite having adequate protective antibodies, a sufficiently large exposure to the toxin can overwhelm the developed response." This is not unique to toxoids and can also occur with viruses or other pathogens.

We have updated the text.

Line 104-108. "While a study showed no significant differences in pertussis and tetanus antibodies in children, another study showed a Th2-biased immune response and a decreased Th1 response, which usually mediates acute infections[35]. In addition, a study in healthcare workers vaccinated against hepatitis B showed high levels of non-responsiveness [35]." The authors base a large part of their review on other review articles. Reference 35 concerns the cost of vaccine development. The authors should focus their review on primary research articles.

There were mistakes in citations in this section. We have updated the references.

Figure 2. "Neutralizing profiles can then be identified." The authors should explain how steps 1 – 9 lead to neutralizing profiles.

We have updated the text and figure legend.

119 - . "The new vaccine platforms (e.g., mRNA) and the COVID-119 19 pandemic provide a unique opportunity to characterize memory in greater detail and 120 identify improved metrics for vaccine durability." It is not clear why the mRNA vaccines would provide a unique opportunity compared with more traditional vaccine approaches.

We have clarified this statement in the text.

Line 250. "In peptide vaccines such as measles/mumps and hepatitis". Measles/mumps is a live attenuated vaccine.

We did not mean the approved vaccine. We were discussing a study with vaccine candidates. We have clarified this in the text.

Figure 3. It is not clear how epitope mapping of antibodies from patients immunized with subunit or killed vaccines can lead to modifications of the vaccine to improve the humoral immune response.

Section 3.3.3 details this point.

Line 448. Paragraph 4.3 has little to do with epidemiology or "Epidemiological advancement".

We have added to the section for clarification.

Reviewer #2

We appreciate the kind review.

Reviewer #3

We appreciate the kind review.

Reviewer #4

However, Figure 1: It would be great if the authors discuss the rationale and importance/limitations of multiple boosters, e.g., 3 or 4 doses.

We appreciate the kind review. We have added more information on the limitation of boosters.

Reviewer 2 Report

I think this is an excellent review . I have been in the vaccine field now for some 45 years and even though significant progress has been made we still lack methodology to screen for T-cells responses, because these are the guys that will do the clean-up.  I was happy with the HLA chapters something that has rendered Noble prize and has not been discussed in the case of COVID (sic). I was very suprised with the low titers of the protective antibodies against the Influenza. This might have to do with the use of cell lines for measuring protective antibodies. I think this would have been an intersting block for the review. In short , congratulation, and I will use this review in my teaching!

Author Response

We appreciate the kind review. 

Reviewer 3 Report

In this exceptionally well-written timely review, the authors discuss the new technological strategies that go beyond measuring the titers to develop safe and efficacious vaccines thus boosting consumer vaccine confidence.  The authors cover a wide array of techniques including epitope binning, HLA typing, and B-cell sequencing in detail with effective figures.

I recommend that this review manuscript be accepted for publication in Vaccines.

Author Response

We appreciate the kind review. 

Reviewer 4 Report

Brooks et al., have submitted the review entitled “Moving beyond titers”. This is an excellent review. I appreciate the author’s didactic description of vaccine development and the major points to consider. Information on technologies is very informative. Although difficult, several parameters have to be considered for a successful vaccine candidate development.  

This reviewer does not have any suggestions.

However, Figure 1: It would be great if the authors discuss the rationale and importance/limitations of multiple boosters, e.g., 3 or 4 doses.

Author Response

We appreciate the kind review. We have added more information on the limitation of boosters.

Round 2

Reviewer 1 Report

The authors did not substantially change their review.